# Research on Six-Wheel Distributed Unmanned Vehicle Path Tracking Strategy Based on Hierarchical Control

**DOI:** 10.3390/biomimetics7040238

**Published:** 2022-12-12

**Authors:** Teng’an Zou, Yulong You, Hao Meng, Yukang Chang

**Affiliations:** 1College of Intelligent Science, National University of Defense Technology, Changsha 410073, China; 2College of Automation Engineering, Nanjing University of Aeronautics and Astronautics, Nanjing 211106, China; 3Chongqing Chang’an Wangjiang Industry Company Limited, Chongqing 404135, China

**Keywords:** distributed unmanned vehicle, hierarchical control, MPC, deterministic torque

## Abstract

For the multi-objective control problem of tracking effect and vehicle stability in the path tracking process of six-wheel distributed unmanned vehicles, a control strategy based on hierarchical control (HC) theory is proposed. A hierarchical kinematic model is designed considering the structural advantages of independent steering and independent driving of the unmanned vehicle, and this model is applied to the path tracking strategy. The strategy is divided into two levels of control. The upper level of control is to use the upper-level kinematic model as the prediction model of model predictive control (MPC), and to convert the solution problem of future control increments into the optimal solution problem of quadratic programming by setting the optimal objective function and constraints. The lower level of control is to map the optimal control quantities obtained from the upper level control to the six-wheel speeds and the four-wheel turning angles through the lower-level kinematics, and to design the six-wheel torque distribution rules based on deterministic torque and stability-based slip rate control for executing the control requirements of the upper level controller to prevent the unmanned vehicle from generating sideslip and precisely generating transverse moment to ensure the stable driving of the unmanned vehicle. Experiments were conducted on the Trucksim/Simulink simulation platform for a variety of road conditions, and the results showed that hierarchical control improved the accuracy of tracking the desired path and the driving stability on complex road surfaces more than MPC.

## 1. Introduction

Unmanned ground vehicles (UGV) are designed with a distributed chassis structure with six-wheel independent drive and four-wheel independent steering (6WID/4WIS) as a platform, with the advantages of fast, flexible and accurate response [1,2,3,4]. Applying it to path tracking control not only improves the accurate tracking of the desired path, but also helps improve the operability and flexibility of unmanned vehicles [5,6]. Scholars at home and abroad have carried out much research on the path tracking control of UGV. The commonly used path tracking control algorithms include PID [7,8], fuzzy control [9,10], model predictive control [11,12,13] and so on.

Zhao et al. have achieved acceptable tracking results based on the PID algorithm for tracking the target heading angle and target position from the path planning level [14,15]. However, the PID control algorithm has several problems: first, the workload of debugging PID control parameters is large; second, the PID control algorithm serves to correct the existing control deviation, and it is difficult to achieve accurate control for unmanned vehicles with large inertia. Zhang et al. proposed an unmanned path-tracking strategy based on fuzzy control and PID. The control strategy uses the fuzzy control theory to determine the desired front wheel heading angle based on lateral deviation, heading deviation and speed, effectively improving the stability of vehicle running and tracking accuracy [16]. You [17] established a pre-target path tracker based on an expert PID algorithm and optimal pre-target control theory, and verified through simulation that the strategy has adequate reliability on a good road surface, but the tracking error is larger on a complex road surface or in low adhesion coefficient road conditions.

Considering the limitations of the PID, many scholars have carried out research on path tracking control based on MPC theory. As a popular feedback control algorithm, MPC is widely used in the industrial production control process [18]. The MPC algorithm is a control method that considers the inputs, outputs and future states of the system and compares them with future reference signals when calculating the optimal control inputs of the system. Given the control signal delay problem existing in PID, MPC adds a prediction model to the control system to predict the future state of the robot, thus avoiding the phenomenon of control lag in PID [19]. Gong and Sun used an integral front-wheel steering vehicle as a research object, and applied kinematic and dynamic models to build a prediction model and performed a quadratic programming solution to improve the accuracy of UGV path tracking on good road surfaces [20,21]. Zhao [22] proposed a preview time adaptive control method for path tracking based on the MPC algorithm and vehicle dynamics model, and used a differential torque control method based on reference heading angle to improve the reliability and accuracy of path tracking under constant torque demand. Wang [23] designed a combined steering and braking path tracking controller based on the MPC algorithm. By linearizing the nonlinear vehicle and tire models, a linear time-varying MPC was designed to improve the real-time performance of the system. Simulation results show that the overall control method had better path tracking performance, less interference between steering and braking, and less influence on the longitudinal motion. Jiang [24,25] proposed a model-free predictive control strategy based on particle swarm optimization (PSO). First, the MFAPC control scheme was improved by integrating vehicle state parameters; then, the main parameters of the improved control scheme were optimized based on the PSO algorithm; finally, the effectiveness of the method under different operating conditions was verified by simulation. Kang [26] designed a six-wheel distributed unmanned vehicle drive control algorithm based on slip steering to determine the torque command on each wheel by determining the optimal longitudinal tire force to keep the slip rate value below the limit and to track the desired tire force. Simulations and tests show that the control method has adequate control performance. Xu [27] proposed an adaptive trajectory tracking control method based on pre-targeting time, which introduces a weighting factor method for differential steering and autonomous steering to achieve coordinated trajectory tracking control for distributed unmanned vehicles with differential and autonomous steering.

Among the available research materials on UGV path-following strategies, the following issues have not been addressed.

(1)Most researchers have only focused on the improvement of accuracy performance during UGV path tracking control, but ignored the problem of stability during vehicle driving.(2)The application of the MPC algorithm for path tracking control causes the problems of long adjustment time and weak anti-interference capability because its calculation is complex and time-consuming, and these problems have not been better solved.(3)The state quantities considered by most researchers based on the MPC algorithm are mainly the position and heading angle of the vehicle, which fail to consider and constrain the fluctuation of slip rate and excessive transverse moment caused by the complex road surface, resulting in the instability phenomenon of UGV.(4)At present, most scholars mainly study the traditional front-wheel steering or four-wheel independent drive unmanned vehicle—addressing the accuracy and stability issues during path tracking of distributed unmanned ground vehicles (DUGV) with six-wheel independent steering and four-wheel independent steering is not common.

For the multi-objective control problem of tracking accuracy and vehicle stability during DUGV path tracking, a distributed unmanned vehicle coordination control strategy is proposed based on hierarchical control theory. The main components of this work are as follows. (1) A hierarchical kinematic model is proposed based on the advantages of independent driving and independent steering of the six-wheel DUGV. (2) Based on MPC theory, the optimal velocity and angle control quantities are calculated by converting the solution problem of future control increments into the optimal solution problem of quadratic programming by setting the optimal objective function and constraints. (3) We designed the torque distribution rules based on deterministic torque and stability-based slip rate, and (4) conducted Simulink/Trucksim simulations to verify the effectiveness and feasibility of the proposed control strategy for path tracking control of the six-wheel DUGV. This paper is organized as follows: Section 2 introduces the hierarchical kinematic model of path tracking for the 6WID robot; Section 3 proposes a path tracking control strategy based on hierarchical control theory; Section 4 is a simulation test to verify the accuracy and reliability of path tracking of the unmanned vehicle under different working conditions; finally, the contributions of this work are summarized in Section 5.

## 2. Mathematical Model

The six-wheeled distributed unmanned vehicle is designed with a two-layer path-tracking kinematic model due to the advantages of independent driving and independent steering. The upper kinematic model does not consider the function of independent steering and independent driving of the unmanned vehicle, and only its center-of-mass velocity and front axle center turning angle are analyzed. The lower kinematic model is based on the six-wheel Ackermann steering theory, which maps the center-of-mass velocity and front axle center turning angle to the wheel velocity and turning angle of each of the six wheels to achieve path motion tracking control for distributed unmanned vehicles. The six-wheel distributed unmanned vehicle motion model is shown in Figure 1.

### 2.1. Upper Layer Kinematic Model

As shown in Figure 2, in the inertial coordinate system OXY, (*X_G_*, *Y_G_*) is the center-of-mass (CoM) coordinate, *φ* is the heading angle of the unmanned vehicle, *θ* is the front axle center rotation angle of the unmanned vehicle, V is the CoM velocity of the unmanned vehicle, L is the axis distance and R is the steering radius. The upper kinematic model of the six-wheeled distributed unmanned vehicle is shown in Equation (1).
(1)$$\left[\begin{array}{c} {\dot{X}}_{G}\\ {\dot{Y}}_{G}\\ \dot{\varphi } \end{array}\right]=\left[\begin{array}{c} V\cos\varphi \\ V\sin\varphi \\ \omega \end{array}\right]=V\left[\begin{array}{c} \cos\varphi \\ \sin\varphi \\ 0 \end{array}\right]+V\frac{\tan\theta }{a}\left[\begin{array}{c} 0\\ 0\\ 1 \end{array}\right]$$

### 2.2. Lower Kinematic Model

The relationship between the control quantities V and θ about the state quantities [X_G_, Y_G_, φ] of the six-wheel distributed unmanned vehicle in the path tracking control process is known by the upper-level kinematics. In the lower kinematics, a six-wheel Ackermann steering model is designed based on the two-wheel Ackermann steering model and the function of the distributed unmanned vehicle with six-wheel independent drive and independent steering [28,29]. The model ensures that each wheel makes a circular motion around the same instantaneous center when steering, so that the wheels are in a pure rolling and no-slip state with the ground, to achieve a smooth turn with a smaller steering radius for unmanned vehicles and to maintain better velocity stability and mechanical response characteristics.

Six-wheel Ackermann steering is shown in Figure 1, where B is the wheelbase, L is the axis distance between the front and rear axles and a is the distance from the front axle to the CoM. $R_{L1}$, $R_{L2}$, $R_{L3}$, $R_{R1}$, $R_{R2}$ and $R_{R3}$ are the steering radii of each wheel center around the center of rotation O. $V_{L1}$, $V_{L2}$, $V_{L3}$, $V_{R1}$, $V_{R2}$, and $V_{R3}$,are the longitudinal velocities of each wheel. $\alpha_{L1}$, $\alpha_{L3}$, $\beta_{L1}$ and $\beta_{L3}$ are the steering angles of the front and rear axle wheels, respectively. From the above geometric relationships, it can be shown that:(2)$$\begin{array}{l} R=\frac{a}{\tan\theta }\\ R_{L1}=\sqrt{a^{2}+{\left(R+\frac{B}{2}\right)}^{2}},R_{L1}=\sqrt{a^{2}+{\left(R-\frac{B}{2}\right)}^{2}}\\ R_{L2}=(R+\frac{B}{2}),R_{L2}=(R-\frac{B}{2})\;\\ R_{L3}=\sqrt{b^{2}+{\left(R+\frac{B}{2}\right)}^{2}},R_{L1}=\sqrt{b^{2}+{\left(R-\frac{B}{2}\right)}^{2}} \end{array}$$
(3)$$\begin{array}{c} \tan\alpha_{L1}=a/(R_{1}+\frac{B}{2}),\tan\alpha_{L3}=b/(R_{2}+\frac{B}{2})\\ \tan\beta_{L1}=a/(R_{3}-\frac{B}{2}),\tan\beta_{L3}=b/(R_{4}-\frac{B}{2}) \end{array}$$

From the instantaneous center theorem, it follows that:(4)$$\frac{V}{R}=\frac{V_{L1}}{R_{L1}}=\frac{V_{L2}}{R_{L2}}=\frac{V_{L3}}{R_{L3}}=\frac{V_{R1}}{R_{R1}}=\frac{V_{R2}}{R_{R2}}=\frac{V_{R3}}{R_{R3}}$$

Equations (2) and (3) are substituted into Equation (4) to obtain the relationship between the longitudinal velocity $V_{i}$ of each wheel about the vehicle velocity V and the front axle center rotation angle θ.
(5)$$\begin{array}{l} V_{L1}=\frac{V\sqrt{a^{2}+{\left(R+\frac{B}{2}\right)}^{2}}}{R}=\frac{V\sqrt{a^{2}+{\left(R+\frac{B}{2}\right)}^{2}}}{\frac{a}{\tan\theta }}\\ V_{R1}=\frac{V\sqrt{a^{2}+{\left(R-\frac{B}{2}\right)}^{2}}}{R}=\frac{V\sqrt{a^{2}+{\left(R-\frac{B}{2}\right)}^{2}}}{\frac{a}{\tan\theta }}\\ V_{L2}=\frac{V(R+\frac{B}{2})}{R}=\frac{V(\frac{a}{\tan\theta }+\frac{B}{2})}{\frac{a}{\tan\theta }} \end{array}\;\begin{array}{l} V_{R2}=\frac{V(R-\frac{B}{2})}{R}=\frac{V(\frac{a}{\tan\theta }-\frac{B}{2})}{\frac{a}{\tan\theta }}\\ V_{L3}=\frac{V\sqrt{b^{2}+{\left(R+\frac{B}{2}\right)}^{2}}}{R}=\frac{V\sqrt{b^{2}+{\left(R+\frac{B}{2}\right)}^{2}}}{\frac{a}{\tan\theta }}\\ V_{R3}=\frac{V\sqrt{b^{2}+{\left(R-\frac{B}{2}\right)}^{2}}}{R}=\frac{V\sqrt{b^{2}+{\left(R-\frac{B}{2}\right)}^{2}}}{\frac{a}{\tan\theta }} \end{array}$$

Based on the above kinematic analysis, the model can realize the mapping of vehicle velocity V and front axle turning angle θ into wheel velocities $\left[V_{L1},\;V_{L2},\;V_{L3},\;V_{R1},\;V_{R2},\;V_{R3}\right]$ and turning angles $\left[\alpha_{L1},\alpha_{L3},\;\beta_{R1},\;\beta_{R3}\right]$ to realize the steering control of a six-wheel distributed unmanned vehicle.

## 3. Control Design

Based on HC theory and hierarchical kinematic model [30,31,32,33], we propose a distributed unmanned ground vehicle coordination control strategy, which is divided into two layers. In the upper layer control, the upper layer kinematic model is used as the prediction model of MPC, and the solution problem of future control increment is converted to the optimal solution problem of quadratic programming by setting the optimal objective function and constraint conditions. In the lower layer control, the optimal control quantity obtained from the upper layer control is mapped to the velocity of six-wheel and four-wheel turning angles through the lower layer kinematics, and the torque distribution rules based on deterministic torque and stability slip rate are designed to execute the control demand calculated by the upper layer controller to protect the unmanned vehicle from sideslip and precisely generate the demand transverse moment to ensure the stability of the unmanned vehicle driving. The distributed unmanned ground vehicle coordination control strategy is shown in Figure 2.

### 3.1. Coordinated Control Strategies for Upper Level Design

The MPC controller can predict the future state of the vehicle by establishing a six-wheeled distributed prediction model of the unmanned vehicle and introducing these state quantities into the QP optimization, so as to obtain the optimal control quantity of the unmanned vehicle trajectory tracking [34].

#### 3.1.1. Six-Wheeled Distributed Unmanned Vehicle Prediction Equation

In this paper, the upper kinematics equation of the six-wheel distributed unmanned vehicle is used as the prediction model. According to Formula (1), the model is defined as a control system with input quantity as $u\left(v,\theta \right)$ and state quantity as $\chi \left(x,y,\varphi \right)$, and its form is Equation (2):(6)$$\dot{\chi }=f(\chi ,u)$$

Each trajectory point on the reference trajectory also satisfies the upper kinematic equation, and its form is Equation (7):(7)$${\dot{\chi }}_{ref}=f(\chi_{ref},u_{ref})$$
where $\chi_{ref}={\left[\begin{array}{ccc} x_{ref} & y_{ref} & \varphi_{ref} \end{array}\right]}^{T}$, $u_{ref}={\left[\begin{array}{cc} v_{ref} & \theta_{ref} \end{array}\right]}^{T}$ and r represents the reference volume.

Expanding Equation (6) at the reference trajectory point using the Taylor series and neglecting higher order terms, we obtain:(8)$$\dot{\chi }=f(\chi_{ref},u_{ref})+\frac{\partial f(\chi ,u)}{\partial \chi }\left(\chi -\chi_{ref}\right)+\frac{\partial f(\chi ,u)}{\partial u}\left(u-u_{ref}\right)$$

The linearized six-wheel distributed unmanned vehicle error model is obtained by subtracting Equation (8) from Equation (7).
(9)$$\dot{\tilde{\chi }}=\chi -\chi_{ref}=\left[\begin{array}{c} \dot{x}\\ \dot{y}\\ \dot{\varphi } \end{array}\begin{array}{c} {\dot{x}}_{ref}\\ {\dot{y}}_{ref}\\ {\dot{\varphi }}_{ref} \end{array}\right]=\left[\begin{array}{ccc} 0 & 0 & -v_{ref}\sin\varphi_{ref}\\ 0 & 0 & v_{ref}\cos\varphi_{ref}\\ 0 & 0 & 0 \end{array}\right]\left[\begin{array}{c} x\\ y\\ \varphi \end{array}\begin{array}{c} x_{ref}\\ y_{ref}\\ \varphi_{ref} \end{array}\right]+\left[\begin{array}{c} \begin{array}{c} \cos\varphi_{ref}\\ \sin\varphi_{ref} \end{array}\\ \frac{\tan\theta }{a} \end{array}\begin{array}{c} \begin{array}{c} 0\\ 0 \end{array}\\ \frac{V\sec^{2}\theta }{a} \end{array}\right]$$

A discrete linearized state space system is obtained by Euler discretization of the preceding term in Equation (9). The system is based on upper-level kinematics, the basis for designing a six-wheel distributed unmanned vehicle path tracking control strategy based on a linear time-varying model predictive control algorithm [35].
(10)$$\tilde{\chi }\left(k+1\right)=A_{k,t}\tilde{\chi }\left(k\right)+B_{k,t}\tilde{u}\left(k\right)$$
where $A_{k,t}=\left[\begin{array}{ccc} 1 & 0 & -v_{ref}\sin\varphi_{ref}T\\ 0 & 1 & v_{ref}\cos\varphi_{ref}T\\ 0 & 0 & 1 \end{array}\right]$, $B_{k,t}=\left[\begin{array}{cc} \cos\varphi_{ref}T & 0\\ \sin\varphi_{ref}T & 0\\ \frac{\tan\theta_{ref}T}{a} & \frac{v_{ref}\sec^{2}\theta_{ref}T}{a} \end{array}\right]$ and T is the sampling period.

Convert Equation (10) as follows:(11)$$\zeta \left(k\left|t\right.\right)=\left[\begin{array}{c} \tilde{\chi }(k\left|t\right.)\\ \tilde{u}(k-1\left|t\right.) \end{array}\right]$$
to obtain a new state space expression:(12)$$\zeta \left(k+1\left|t\right.\right)={\tilde{A}}_{k,t}\zeta \left(k\left|t\right.\right)+{\tilde{B}}_{k,t}∆u(k\left|t\right.)$$
where ${\tilde{A}}_{k,t}=\left[\begin{array}{cc} A_{k,t} & B_{k,t}\\ 0_{m\times n} & I_{m} \end{array}\right]$, ${\tilde{B}}_{k,t}=\left[\begin{array}{c} B_{k,t}\\ I_{m} \end{array}\right]$, n is the state volume dimension and m is the control volume dimension.

At moment t, the state space equation for the future $N_{p}$ steps of the system is
(13)$$\begin{array}{l} \zeta \left(t+1\right)={\tilde{A}}_{k}\zeta \left(t\right)+{\tilde{B}}_{k}∆u(t)\\ \zeta \left(t+2\right)={\tilde{A}}_{k}\zeta \left(t+1\right)+{\tilde{B}}_{k}∆U(t+1)={\tilde{A}}_{k}\left({\tilde{A}}_{k}\zeta \left(t\right)+{\tilde{B}}_{k}∆u(t)\right)\\ \;\;\;={\tilde{A}}_{k}{}^{2}\zeta \left(t\right)+{\tilde{A}}_{k}{\tilde{B}}_{k}∆u(t)+{\tilde{B}}_{k}∆u(t+1)\\ \zeta \left(t+3\right)={\tilde{A}}_{k}{}^{3}\zeta \left(t\right)+{\tilde{A}}_{k}{}^{2}{\tilde{B}}_{k}∆u(t)+{\tilde{A}}_{k}{\tilde{B}}_{k}∆u(t+1)+{\tilde{B}}_{k}∆u(t+2)\\ \vdots \\ \zeta \left(t+N_{c}\right)={\tilde{A}}_{k}{}^{N_{c}}\zeta \left(t\right)+{\tilde{A}}_{k}{}^{N_{c}-1}{\tilde{B}}_{k}∆u(t)+\cdots +{\tilde{A}}_{k}{\tilde{B}}_{k}∆u(t+N_{c}-2)+{\tilde{B}}_{k}∆u(t+N_{c}-1)\\ \vdots \\ \zeta \left(t+N_{p}\right)={\tilde{A}}_{k}{}^{N_{p}}\zeta \left(t\right)+{\tilde{A}}_{k}{}^{N_{p}-1}{\tilde{B}}_{k}∆u(t)+\cdots +{\tilde{A}}_{k}{\tilde{B}}_{k}∆u(t+N_{p}-2)+{\tilde{B}}_{k}∆u(t+N_{p}-1) \end{array}$$
where $N_{p}$ is the prediction time domain and $N_{c}$ is the control time domain; $N_{p}>N_{c}$.

At time t, the trajectory state equation of the control system in the predicted time domain is as follows:(14)$$Y\left(t\right)={Μ}_{t}\zeta \left(t\right)+\Omega_{t}\Delta U\left(t\right)$$
where $Y\left(t\right)=\left[\begin{array}{c} \zeta \left(t+1\left|t\right.\right)\\ \zeta \left(t+2\left|t\right.\right)\\ \cdots \\ \zeta \left(t+N_{C}\left|t\right.\right)\\ \cdots \\ \zeta \left(t+N_{P}\left|t\right.\right) \end{array}\right]$, ${Μ}_{t}=\left[\begin{array}{c} {\tilde{A}}_{k}\\ {\tilde{A}}_{k}{}^{2}\\ \cdots \\ {\tilde{A}}_{k}{}^{N_{c}}\\ \cdots \\ {\tilde{A}}_{k}{}^{N_{p}} \end{array}\right]$, $\Omega_{t}=\left[\begin{array}{cccc} {\tilde{B}}_{k} & 0 & 0 & 0\\ {\tilde{A}}_{k} & {\tilde{B}}_{k} & 0 & 0\\ \cdots & \cdots & \ddots & \cdots \\ {\tilde{A}}_{k}{}^{N_{c}}{\tilde{B}}_{k} & {\tilde{A}}_{k}{}^{N_{c}-1}{\tilde{B}}_{k} & \cdots & {\tilde{B}}_{k}\\ \vdots & \vdots & \ddots & \vdots \\ {\tilde{A}}_{k}{}^{N_{p}}{\tilde{B}}_{k} & {\tilde{A}}_{k}{}^{N_{p}-1}{\tilde{B}}_{k} & \cdots & {\tilde{B}}_{k} \end{array}\right]$ and $∆U=\left[\begin{array}{c} ∆u(t)\\ ∆u(t+1)\\ \vdots \\ ∆u(t+N_{c}-1)\\ \vdots \\ ∆u(t+N_{p}-1) \end{array}\right]$.

The analysis of the prediction expression (14) shows that the state quantity $\zeta \left(k\left|t\right.\right)$ at the current moment and the control increment ∆U in the predicted time domain can be used to calculate the state quantity and the control output in the predicted time domain to overcome the control delay problem in the PID algorithm.

#### 3.1.2. QP Optimization

∆U in the prediction expression (14) is unknown, and can only be obtained by setting an appropriate objective function and optimizing the solution. To ensure that the distributed unmanned vehicle can track the desired target path quickly and accurately, we designed an objective function that optimizes the three parameters: state quantity deviation, control quantity and control increment [36].
(15)$$J\left(\zeta \left(t\right),u\left(t-1\right),\Delta u\left(t\right)\right)={\sum_{i=1}^{N_{p}}\left\|\zeta \left(t+i\left|t\right.\right)-\zeta_{ref}\left(t+i\left|t\right.\right)\right\|}_{Q}^{2}+{\sum_{i=1}^{N_{c}}\left\|∆u\left(t\right)\right\|}_{R}^{2}+\rho \varepsilon^{2}$$
where ρ is the weighting factor and ε is the relaxation factor.

The first term reflects the ability of the system to track the reference trajectory, and the second term is the requirement for the control quantity to change smoothly. Q and R are the weight matrices. The whole objective function is to enable the robot to track the desired trajectory quickly and smoothly. In addition, in the robot control system, the control quantity, control increment, and state quantity deviation need to be constrained with the following constraints.
(16)$$\begin{array}{l} u_{\min}\left(t+k\right)\leq u\left(t+k\right)\leq u_{\max}\left(t+k\right)\\ \Delta u_{\min}\left(t+k\right)\leq \Delta u\left(t+k\right)\leq \Delta u_{\max}\left(t+k\right)\\ y_{\min}\left(t+k\right)\leq y\left(t+k\right)\leq y_{\max}\left(t+k\right) \end{array}$$
where $k=0,1,\cdots ,N_{c}-1$.

The corresponding matrix operation is performed on Formula (14) and Formula (15), and the solution problem of constrained optimization in MPC is equivalent to the following quadratic programming problem [37]:(17)$$\underset{\Delta U,\varepsilon }{\min}{\left[\Delta U{\left(t\right)}^{T},\varepsilon \right]}^{T}H_{t}\left[\Delta U{\left(t\right)}^{T},\varepsilon \right]+G_{t}\left[\Delta U{\left(t\right)}^{T},\varepsilon \right]$$
where $H_{t}=\left[\begin{array}{cc} \Omega_{t}{}^{T}Q\Omega_{t}+R & 0\\ 0 & \rho \end{array}\right]$, $G_{t}=\left[\begin{array}{cc} 2e_{t}{}^{T}Q\Omega_{t} & 0 \end{array}\right]$ and $e_{t}$ is the tracking error in the predicted time domain.

After the optimal solution of the quadratic programming of Equation (17) is obtained, a series of control incremental inputs ΔU in the control time domain at future moments will be obtained.
(18)$$\;\Delta U_{t}=\left[∆u_{t},\Delta u_{t+1},\ldots ,\Delta u_{t+N_{c-1}}\right]$$

According to the basic principle of MPC, the first element of this control sequence is applied to the system as an actual control input increment.
(19)$$u\left(t\right)=u\left(t-1\right)+\Delta u$$

Through the lower kinematics, the actual control quantities obtained from Equation (19) can be mapped to the velocity and steering angle control quantities of the respective wheels of the unmanned vehicle, and the expected wheel torque is calculated by the incremental PID controller. In a new control cycle, the system re-predicts the state quantities in the next time domain based on the current state information, and then obtains a new sequence of control increments through an optimization process.

### 3.2. Coordinated Control Strategies for Lower Level Design

In the upper level control of unmanned vehicles, the road state information is not taken into account, so it is not always possible to achieve stable and sufficient tracking results when performing path tracking control on complex road surfaces. Therefore, a control strategy based on deterministic torque and adaptive fuzzy PID-based Acceleration Slip Regulation (ASR) is designed in the lower controller. The strategy achieves precise generation of the required yaw moment for control and stabilization of the wheel slip rate by redistributing the torque on each wheel. It ensures the stability of unmanned vehicle driving while satisfying the upper level control requirements.

#### 3.2.1. Distribution Strategy Based on Deterministic Moments

For the six-wheeled distributed unmanned vehicle to maintain the ability to track the desired path, the ideal method is that the yaw moment should be equal to a determined value to avoid all sideslips of the tires, but this method is only effective at low speeds and on good road conditions, and it is impossible to avoid all vehicle sideslips at high speeds and on complex roads [38].

For the advantages of independent drive and independent steering of distributed unmanned vehicles, a new transverse pendulum moment control strategy is proposed in this paper. This strategy distributes the longitudinal wheel moments in such a way as to satisfy the required transverse sway moments for upper level control while keeping the torque difference between individual wheels to a minimum. Its function is to avoid the excessive transverse moment that may cause the unmanned vehicle to skid or oversteer at high speeds or on complex road surfaces, and to achieve the goals of minimizing the error of path tracking accuracy and maintaining vehicle stability.

According to the six-wheel Ackerman steering principle, the longitudinal force of the wheel is equal to the tangential force. To generate the yaw moment required for control, the following wheel driving moment distribution is calculated.
(20)$$T_{total}=T_{L1}+T_{L2}+T_{L3}+T_{R1}+T_{R2}+T_{R3}=F_{L1}r+F_{L2}r+F_{L3}r+F_{R1}r+F_{R2}r+F_{R3}r$$
where $T_{total}$ is the wheel torque generated by the incremental PID controller after the upper level control, $T_{i}$ is the sum of moments, $F_{i}$ is the driving force of each driving wheel, r is the wheel radius and i=L1, L2 L3, R1, R2 R3.

Six-wheel distributed unmanned vehicle total transverse moment Μ:(21)$$\begin{array}{l} Μ=F_{L1}R_{L1}+F_{L2}R_{L2}+F_{L3}R_{L3}+F_{R1}R_{R1}+F_{R2}R_{R1}+F_{R3}R_{R1}\\ \;\;\;=\frac{T_{L1}}{r}R_{L1}+\frac{T_{L2}}{r}R_{L2}+\frac{T_{L3}}{r}R_{L3}+\frac{T_{R1}}{r}R_{R1}+\frac{T_{R2}}{r}R_{R1}+\frac{T_{R3}}{r}R_{R1} \end{array}$$

Because $T_{total}$ and Μ are related to the driving demand, there are multiple solutions for the driving force distribution. To avoid generating too much demand torque causing the unmanned vehicle to lose control in oversteering, our strategy is designed to minimize the difference in torque at each wheel while satisfying the demand transverse swing torque, minimized by the equation:(22)$$\sum_{i=1}^{6}{\left(T_{i}-\frac{T_{total}}{6}\right)}^{2}$$

This equation is solved using the Lagrange multiplier method [39].
(23)$$L\left(T_{i},\lambda_{i}\right)=\sum_{i=1}^{6}{\left(T_{i}-\frac{T_{total}}{6}\right)}^{2}+\lambda_{1}\left(\sum_{i=1}^{6}T_{i}-T_{total}\right)+\lambda_{2}\left(\sum_{i=1}^{6}R_{i}\frac{T_{i}}{r}-Μ\right)$$

Equation (23) solves the partial differential equation for $T_{i}$:(24)$$\frac{\partial L\left(T_{i},\lambda_{i}\right)}{\partial T_{i}}=2T_{i}+\lambda_{1}+\frac{\lambda_{2}R_{i}}{r}-\frac{T_{total}}{3}=0$$
(25)$$T_{i}=\frac{1}{6}\left(T_{total}-3\lambda_{1}-3\frac{\lambda_{2}R_{i}}{r}\right)$$

Substituting Equation (25) into Equations (21) and (22), we obtain:(26)$$0=\lambda_{1}+\frac{\lambda_{2}\sum R_{i}}{r}$$
(27)$$Μ=\frac{T_{total}}{6r}\sum R_{i}-\frac{\lambda_{1}}{2r}\sum R_{i}-\frac{\lambda_{2}}{2r^{2}}{\left(\sum R_{i}\right)}^{2}$$

λ is obtained from the joint solution of Equations (26) and (27).
(28)$$\lambda_{1}=c_{1}+d_{1}Μ$$
(29)$$\lambda_{2}=c_{2}+d_{2}Μ$$
where $c_{1}=\frac{T_{total}}{6}$, $d_{1}=-\frac{r}{\sum R_{i}}$, $c_{2}=\frac{T_{total}r}{6\sum R_{i}}$ and $d_{2}=-\frac{r^{2}}{{\left(\sum R_{i}\right)}^{2}}$.

Substituting Equations (28) and (29) into Equation (27), $T_{i}$ can be expressed by Μ as
(30)$$T_{i}={Κ}_{i}+\Omega_{i}Μ$$
where ${Κ}_{i}=\frac{1}{6}\left(T_{total}-3c_{1}+\frac{3R_{i}c_{2}}{r}\right)$ and $\Omega_{i}=\frac{R_{i}d_{2}}{2r}-\frac{d_{1}}{2}$.

According to Equation (30), the yaw moment required by the upper control is accurately generated based on the deterministic moment distribution strategy to avoid the instability of the unmanned vehicle caused by over-steering and sideslip.

#### 3.2.2. Adaptive PID-Based Drive Anti-Slip Control

ASR control is a kind of active safety control of automobiles [40]. Its purpose is to obtain the maximum adhesion coefficient by controlling the wheel slip rate, and to avoid the decline in lateral stability caused by excessive wheel slip [41]. The wheel slip rate formula is as follows.
(31)$$S=\frac{V_{x}-V}{V_{x}}$$
where V is vehicle speed and $V_{x}$ is wheel speed. When the wheel rolls purely: $V_{x}=V$, S=0. When the wheel rolls and slides at the same time: $V_{x}>V$, 0<S<100%. The greater the wheel sliding rate, the greater the proportion of the wheel sliding component in the movement.

In this paper, the method of logical threshold value commonly used in traditional vehicles is used to carry out the drive anti-skid control. The main way is to set the drive anti-skid control threshold, and realize the drive anti-skid control by controlling the real-time slip rate of the wheels. The optimal slip rate is set as 20% [42].

We applied fuzzy control theory based on an incremental PID control algorithm to design an adaptive fuzzy PID controller to achieve the tracking control of the optimal slip rate of the wheel. The controller adjusts the proportional, integral and differential parameters in the incremental PID algorithm online by inputting deviations and the rate of change of deviations to improve the controller’s path tracking performance and enhance the controller’s self-adaptive capability. Adaptive fuzzy PID control is mainly composed of three elements, namely fuzzification, fuzzy inference and clarity, with the corresponding elements input affiliation function, development of control rules and logic judgment [43,44], as shown in Figure 3.

The adaptive fuzzy PID controller takes the difference between the target value of the wheel slip rate and the actual slip rate and the rate of change of the difference as the input of the controller, and adaptively adjusts the parameters *K_p_*, *K_i_* and *K_d_* of the incremental PID by the fuzzy inference method. Equation (32) is the adjustment formula, and the parameter adjustment process is shown in Figure 4.
(32)$$k_{p}=k_{p0}+\Delta k_{p},\;k_{i}=k_{i0}+\Delta k_{i},\;k_{d}=k_{d0}+\Delta k_{d}$$
where $k_{p0}$, $k_{i0}$ and $k_{d0}$ are the initial parameters of the incremental PID controller; $\Delta k_{p}$, $\Delta k_{i}$ and $\Delta k_{d}$ are the adjustment parameters of the fuzzy controller output; and $k_{p}$, $k_{i}$ and $k_{d}$ are the final parameters of the adaptive fuzzy PID controller.

## 4. Simulation Results

### 4.1. Distributed Unmanned Vehicle Simulation Platform Based on Trucksim/Simulink

A six-wheeled distributed unmanned vehicle simulation platform with independent drive and independent steering was built jointly with Trucksim and Simulink. The six-wheel distributed unmanned vehicle dynamics model is mainly composed of three parts: body form factor and inertia parameters, vehicle drive/steering system, suspension and tire system. As shown in Figure 5, this paper completes the parameterized models of vehicle form factor, aerodynamics, suspension and tire system in Trucksim, while disconnecting the drive end and steering end interfaces in Trucksim, establishing the drive motor model and steering motor model in Simulink and obtaining the drive torque and steering angle control quantities through the data interfaces in Trucksim.

The data signal interaction with the Trucksim software 2019 was completed in the MATLAB/Simulink environment, integrating six drive motor modules, four steering motor modules, the driver module and the control algorithm module, as shown in Figure 6. The vehicle model parameters are shown in Table 1.

The distributed unmanned vehicle has two-wheel steering, four-wheel steering, four-wheel differential steering, center steering, crabbing, center differential steering and differential steering modes due to its six-wheel independent drive four-wheel independent steering system. The six steering kinematic models are shown in Figure 7.

The above six steering function modes can be realized on the Trucksim/Simulink-based distributed unmanned vehicle simulation platform through wheel speed and turning angle control, as shown in Figure 6, which establishes a simulation platform for a six-wheel independent drive with four-wheel independent steering. Distributed unmanned vehicle path tracking experiments are carried out on the folio and Sine Sweep pavements provided by the simulation platform.

### 4.2. Split Mu Straight Path Following Experiment

Linear trajectory tracking was carried out on the split road surface at a speed of 40 km/h. The adhesion coefficient of the split road surface was 0.2 on the left side and 0.5 on the right side. The simulation animation is shown in Figure 8. Three methods are used for simulation experiment comparison, namely “no control”, “MPC” and “hierarchical control” designed in this paper, where “no control” refers to no path tracking controller, only including speed controller. “MPC” means that there is no lower control, the upper kinematic model is used as the prediction model of MPC and the optimal control quantity is calculated through the model predictive control theory and directly mapped to the speed control quantity of six wheels and the four-wheel angle control quantity.

Simulation results: Shown in Figure 9 is the trajectory tracking curve. In the case of no control, the unmanned vehicle will gradually drift toward the low adhesion coefficient road surface, and the maximum drift is 1.215. Using the MPC algorithm, due to the inconsistent adhesion coefficient on the left and right side, the longitudinal force that the ground can provide to the wheels is inconsistent, the unmanned vehicle has a long adjustment time and an increasingly large drift with the driving time, and it cannot achieve effective tracking and eventually loses control. After adopting coordinated control, the torque distribution of the wheels based on deterministic torque and slip rate control was carried out, and stable and accurate tracking of the target path was achieved. As shown in Figure 10, in terms of speed tracking, the PID and MPC algorithms reached the target velocity at the same time, but both had steady-state errors, while the coordinated control reached the target velocity 2 s faster than MPC, with a faster response time and no steady-state errors. As shown in Figure 11 and Figure 12, the unmanned vehicle using the MPC algorithm has a large fluctuation in the yaw angle and yaw rate, among which the maximum yaw rate fluctuation reaches −19.48 deg/s to 19.27 deg/s. However, after coordinated control, the vehicle’s yaw rate stabilizes between −1.446 deg/s and 1.548 deg/s. This number is already small enough to satisfy the stable running of unmanned vehicles. Therefore, the coordinated control strategy is more beneficial to the safe driving of the vehicle.

To better test the performance of the control system, simulation tests with vehicle speeds of 20 km/h, 30 km/h, 50 km/h and 60 km/h were conducted. The simulation results were as follows.

As shown in Figure 13 and Figure 14, the unmanned vehicle can track the expected speed of 20 km/h at 1.815 s and can accurately track the target trajectory. The unmanned vehicle can track the desired speed of 30 km/h at 2.511 s and can accurately track the target trajectory. The unmanned vehicle can track the expected speed of 50 km/h at 4.298 s, and achieve accurate tracking of the target track when it travels to 158.9 m. The unmanned vehicle can track the expected speed of 60 km/h at 10.14 s, and achieve accurate tracking of the target trajectory when it travels to 282.1 m. The control system of the unmanned vehicle can maintain the slip rate at 20%, as shown in Figure 15. At this time, the wheels can obtain the maximum adhesion coefficient, which is conducive to the stability control of the unmanned vehicle.

### 4.3. Sine Sweep Straight Path Following Experiment

Linear trajectory tracking was performed on a Sine sweep road at a velocity of 40 km/h with a road surface adhesion coefficient of 0.85. The simulation animation is shown in Figure 16.

Simulation results: The trajectory tracking curve is shown in Figure 17. Under no control, the unmanned vehicle will gradually shift, and the maximum shift is 2.194. With the MPC algorithm, the unmanned vehicle has a long adjustment time and the actual running trajectory fluctuates between −0.8028 and 0.06908, which cannot achieve effective tracking. After adopting coordination control, there is a small fluctuation in the tracking trajectory in the first 150 m, and after 150 m, accurate tracking of the target path was achieved. As shown in Figure 18, in terms of velocity tracking, the target speed was reached at 13.91 with PID control; with MPC control, the velocity tracking showed overshoot and the target velocity was reached at 17.78 s; with the coordinated control strategy, the accurate tracking of the target velocity was achieved at 5.137 s. As shown in Figure 19, the yaw angle of the unmanned vehicle fluctuated between −6.454 and 5.956 when the MPC algorithm was used. However, after the coordinated control strategy was used, the heading angle of the unmanned vehicle fluctuated slightly and stabilized at 0° after 12 s. As shown in Figure 20, when the MPC algorithm was used, the yaw rate of the unmanned vehicle fluctuated between −8.76 deg/s and 11.13 deg/s, while when the coordinated control strategy was used, the yaw rate of the unmanned vehicle stabilized between −0.7366 deg/s and 0.8099 deg/s after 10 s.

To better test the performance of the algorithm, simulation tests of the vehicle speeds of 20 km/h, 30 km/h, 50 km/h and 60 km/h were conducted on the Sine sweep straight road.

Simulation results: As shown in Figure 21 and Figure 22, when the expected speed is 20 km/h, the speed can reach the expected speed at 2.407 s, and the maximum offset of the trajectory error is 0.07632 m. When the expected speed is 30 km/h, the speed can reach the expected speed at 2.597 s, and the maximum offset of the trajectory error is 0.0128 m. When the expected speed is 50 km/h, the speed can reach the expected speed by 4.226 s, and the maximum offset of the tracking error is 0.2558 m. When the expected speed is 60 km/h, the speed can reach the expected speed at 4.702 s, and the maximum offset of the tracking error is 0.3383 m. Therefore, under different speed conditions, the unmanned vehicle can stably track the target trajectory.

Figure 23 shows the curve of the six-wheel slip rate at 50 km/h. The control system can make the slip rate of the unmanned vehicle stable at 20% in 2 s.

In addition, through the simulation experiments of different speeds under the Split mu straight road and Sine sweep straight road, it was shown that when the unmanned vehicle used the coordinated control strategy for trajectory tracking at low speeds (such as 20 km/h and 30 km/h), it had suitable path tracking performance and accuracy. With the increase in the target speeds of the unmanned vehicle (such as 50 km/h and 60 km/h), the yaw velocity of the vehicle increases with the increase in the vehicle traveling speed, and the yaw torque required by the unmanned vehicle also increases. Therefore, the coordinated control strategy requires longer control convergence time, resulting in the increase in lateral deviation at the initial moment. However, the coordinated control strategy can still accurately track the desired trajectory in a short time to meet the requirements of unmanned vehicle trajectory tracking.

## 5. Conclusions

(1)Based on the physical structure of distributed unmanned vehicles with six-wheel independent drive and four-wheel independent steering, we designed a hierarchical kinematic model of unmanned vehicle path tracking based on the six-wheel Ackermann theory.(2)Based on HC theory, a coordinated control strategy for path tracking and stability of distributed unmanned vehicles was designed. The strategy is divided into two levels of control. In the upper level of control, the upper kinematic model is used as the prediction model of MPC, and the solution problem of future control increments is converted into the optimal solution problem of quadratic programming by setting the optimal objective function and constraints. The lower level of control is to map the optimal control quantities obtained from the upper level to the six-wheel speed control quantities and the four-wheel turning angle control quantities through the lower-level kinematics, and design the six-wheel torque distribution rules based on deterministic torque and stability-based slip rate control for executing the control demand calculated by the upper-level controller to prevent the unmanned vehicle from producing sideslip and to precisely generate the demand transverse moment to ensure the stability of the unmanned vehicle driving.(3)An unmanned vehicle simulation platform based on Trucksim/Simulink with six-wheel independent drive and four-wheel independent steering was established, and the path tracking tests of 20 km/h, 30 km/h, 40 km/h, 50 km/h and 60 km/h were carried out on this platform on the Split mu straight road and the Sine sweep straight road. The simulation results show that the coordinated control has better response characteristics than MPC, which can output the deterministic moment and control the wheel slip rate at 20%, improving the accuracy and stability of the unmanned vehicle path tracking. Therefore, the coordinated control strategy can achieve stable and accurate path tracking under various working conditions.(4)In the coordinated control algorithm designed in this paper, there is a coupling relationship between speed tracking and path tracking. In future research, the joint control of the speed and path of the unmanned vehicle will be studied. In addition, the parameters of the vehicle model and control system in this paper are taken as fixed parameters. When the vehicle is under different working conditions, these state parameters have errors with the actual values. The future research direction is to further apply the adaptive control technology to the research of this paper, so that it can have the ability of online update and dynamic change.

## Figures and Tables

**Figure 1 biomimetics-07-00238-f001:**
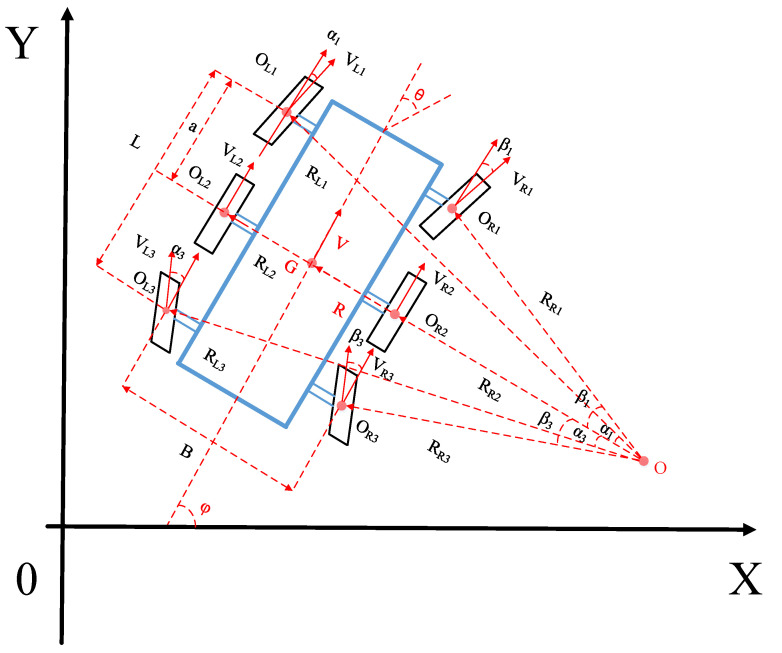
Six-wheel distributed unmanned vehicle path-tracking kinematic model.

**Figure 2 biomimetics-07-00238-f002:**
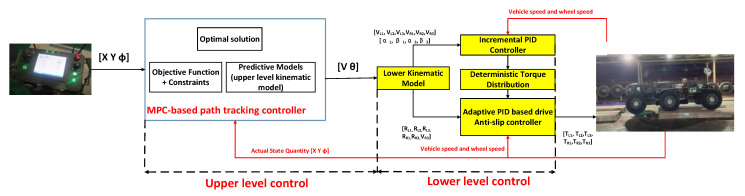
Coordinated control strategy of the distributed unmanned vehicle with a fully electric drive.

**Figure 3 biomimetics-07-00238-f003:**
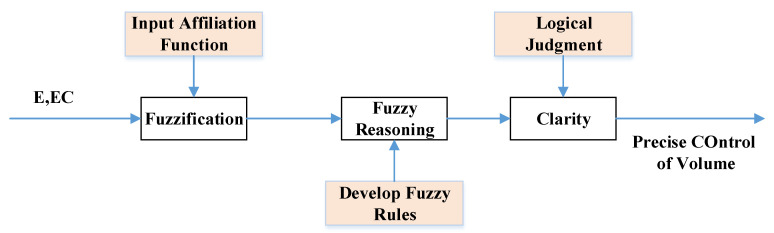
Design a fuzzy controller flow diagram.

**Figure 4 biomimetics-07-00238-f004:**
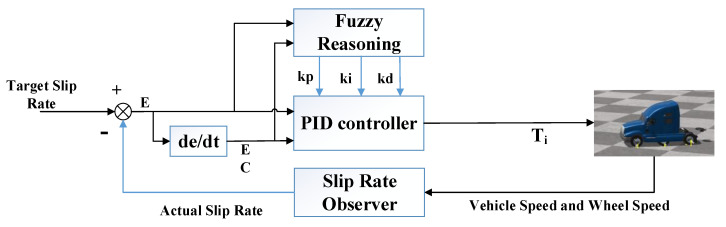
Slip rate controller based on adaptive fuzzy PID.

**Figure 5 biomimetics-07-00238-f005:**
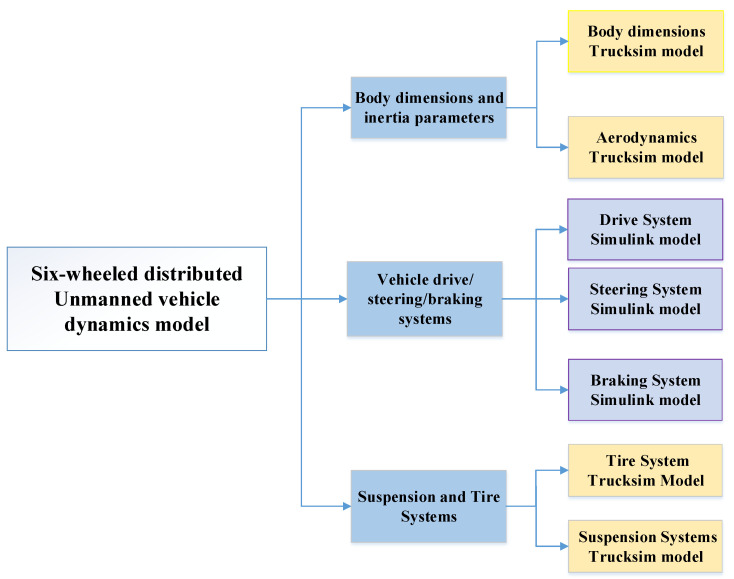
Six-wheeled distributed unmanned vehicle dynamics model.

**Figure 6 biomimetics-07-00238-f006:**
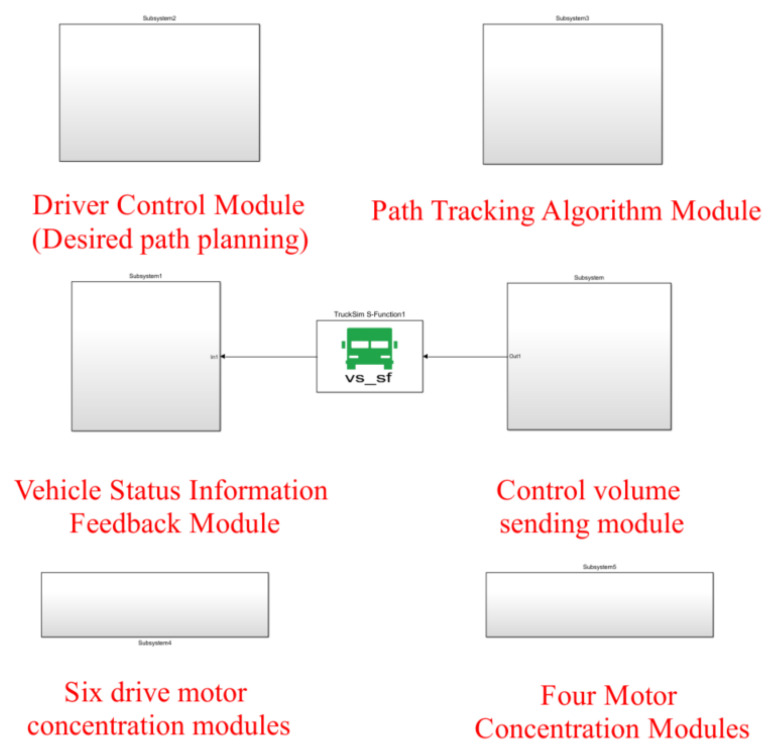
Simulation module under MATLAB/Simulink.

**Figure 7 biomimetics-07-00238-f007:**
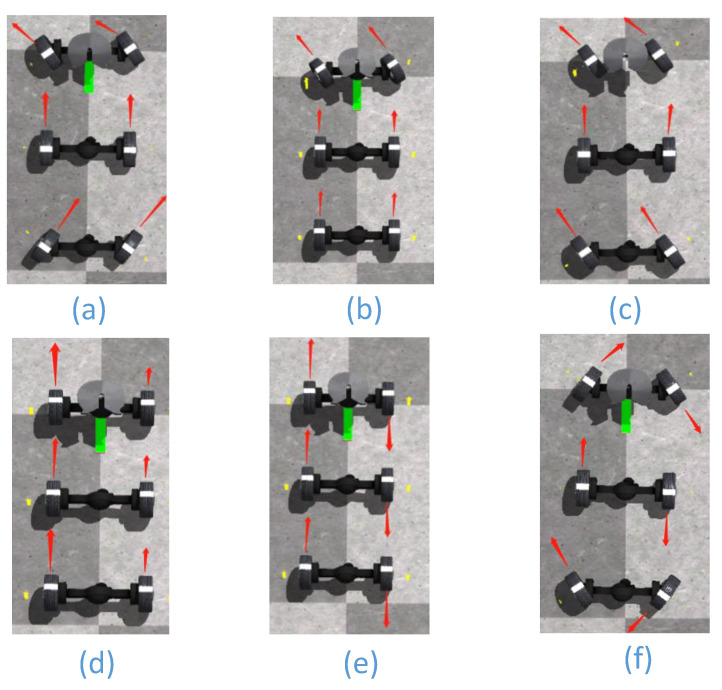
Six steering motion modes of six-wheel distributed unmanned vehicle: (**a**) four-wheel Akerman steering; (**b**) two-wheel Akerman steering; (**c**) crab walking; (**d**) differential steering; (**e**) differential center steering; (**f**) center steering. The direction of the red arrow is the direction of the wheel movement.

**Figure 8 biomimetics-07-00238-f008:**
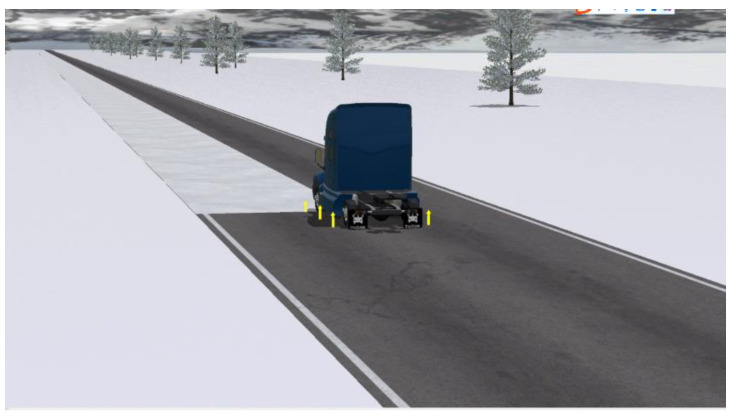
Split mu straight road. The yellow arrow represents the force on the wheel in the z-axis direction.

**Figure 9 biomimetics-07-00238-f009:**
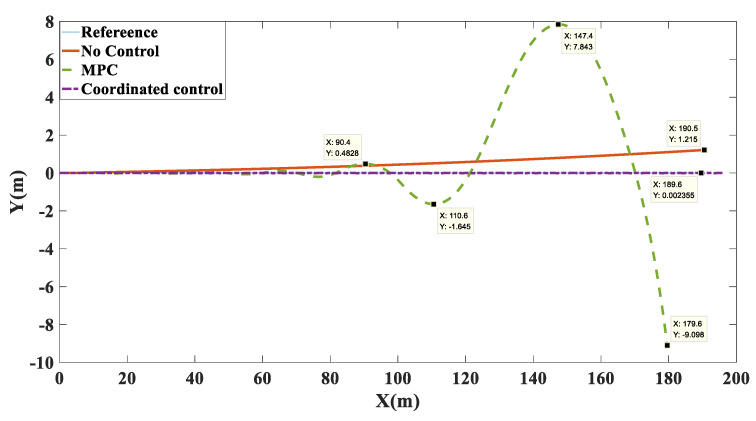
Path tracking comparison chart on the split mu straight road.

**Figure 10 biomimetics-07-00238-f010:**
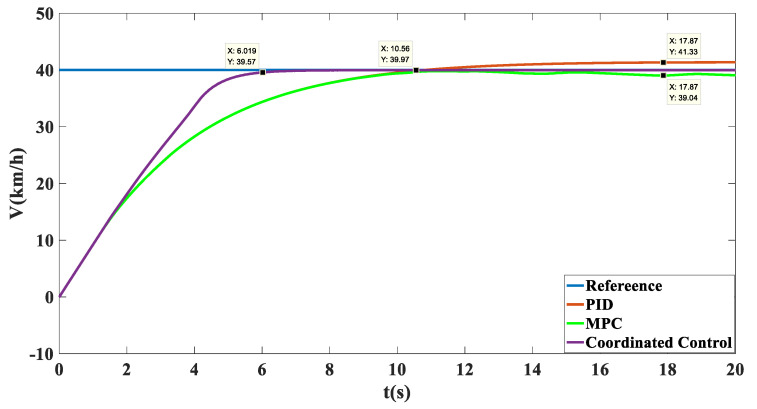
Speed tracking comparison chart on the split mu straight road.

**Figure 11 biomimetics-07-00238-f011:**
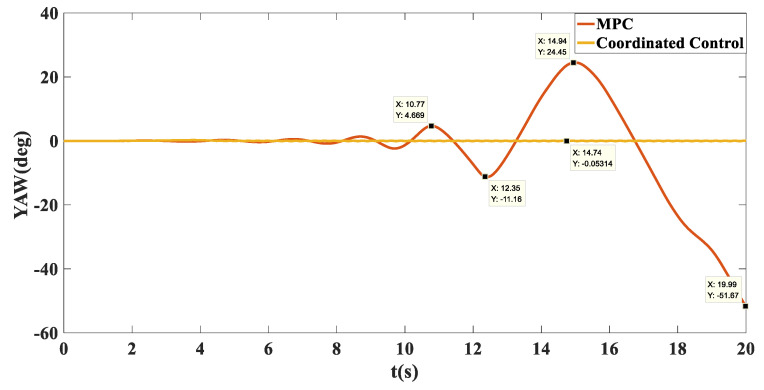
Comparison chart of heading angle tracking on the split mu straight road.

**Figure 12 biomimetics-07-00238-f012:**
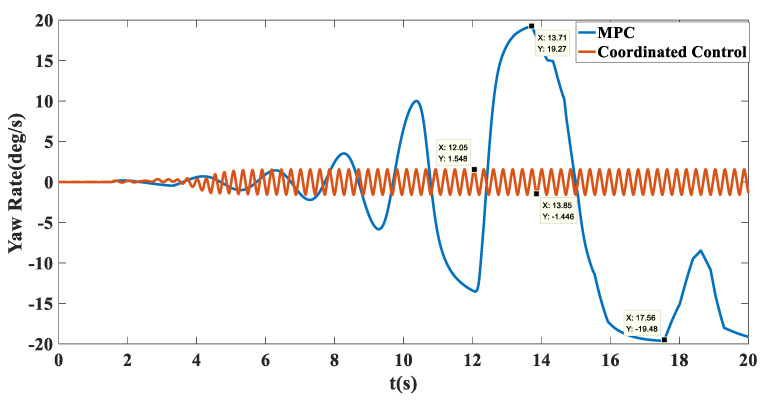
Transverse pendulum angular velocity tracking comparison chart on the split mu straight road.

**Figure 13 biomimetics-07-00238-f013:**
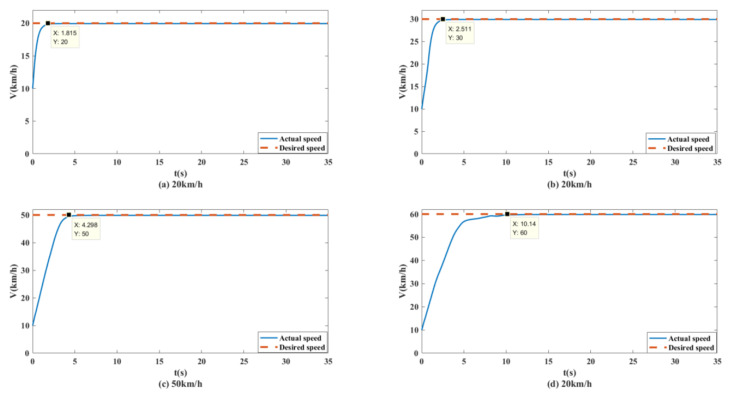
Speed tracking curves of unmanned vehicles at different speeds on the split mu straight road.

**Figure 14 biomimetics-07-00238-f014:**
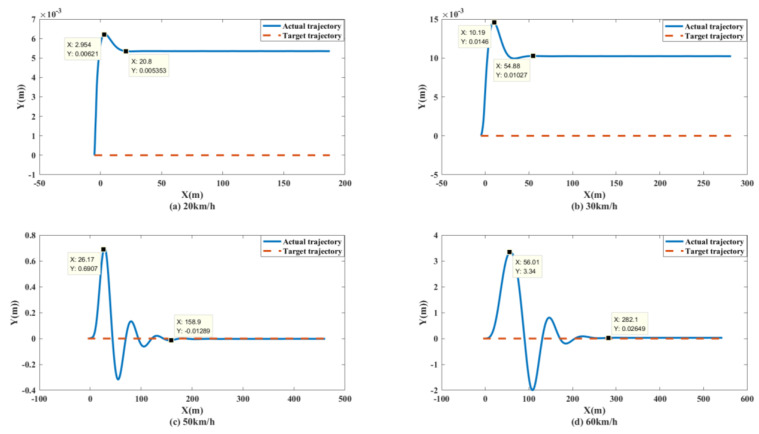
Path tracking curves of unmanned vehicles at different speeds on the split mu straight road.

**Figure 15 biomimetics-07-00238-f015:**
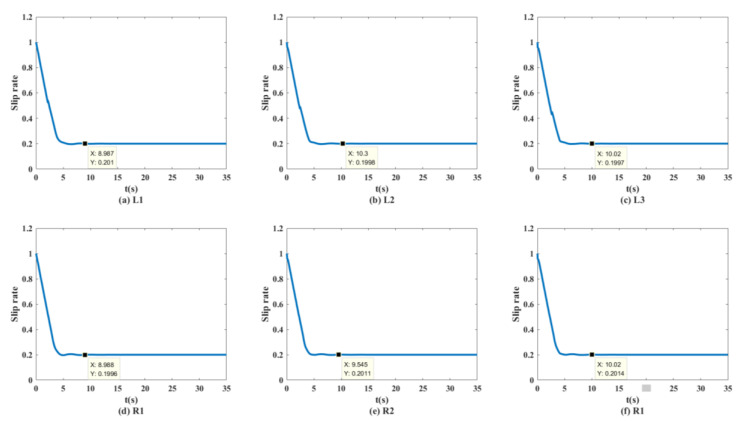
Wheel slip rate of the distributed unmanned vehicle at 50 km/h on the split mu straight road.

**Figure 16 biomimetics-07-00238-f016:**
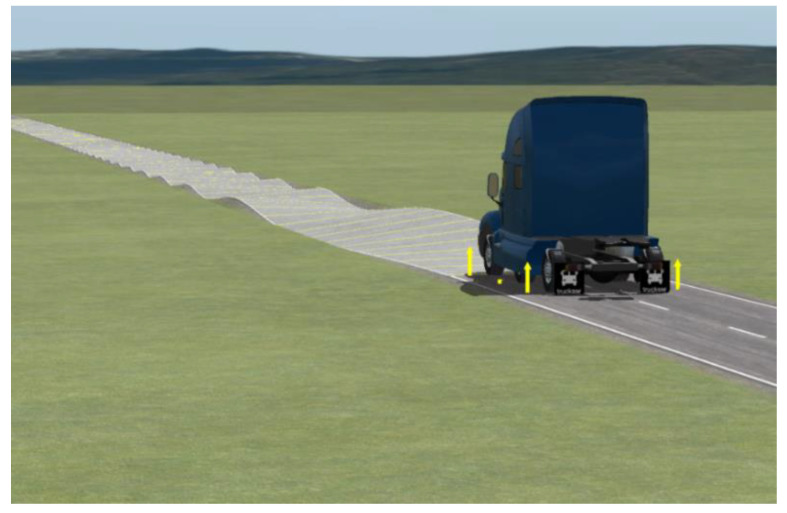
Sine sweep straight road. The yellow arrow represents the force on the wheel in the z-axis direction.

**Figure 17 biomimetics-07-00238-f017:**
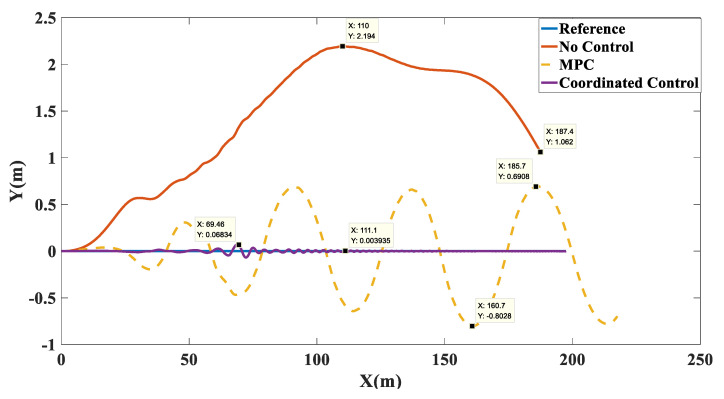
Path tracking comparison chart on the sine sweep straight road.

**Figure 18 biomimetics-07-00238-f018:**
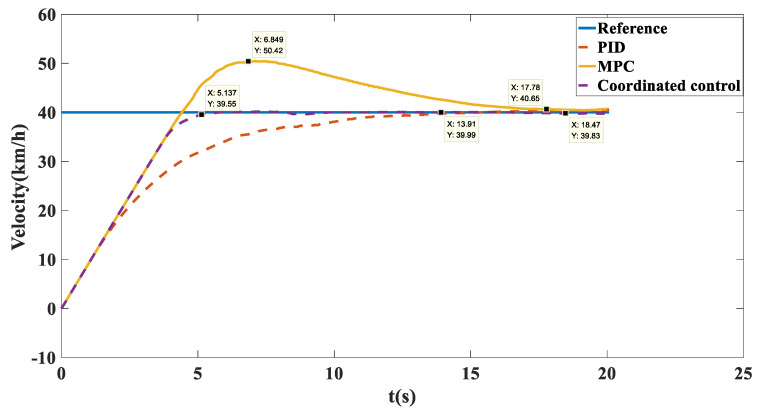
Speed tracking comparison chart on the sine sweep straight road.

**Figure 19 biomimetics-07-00238-f019:**
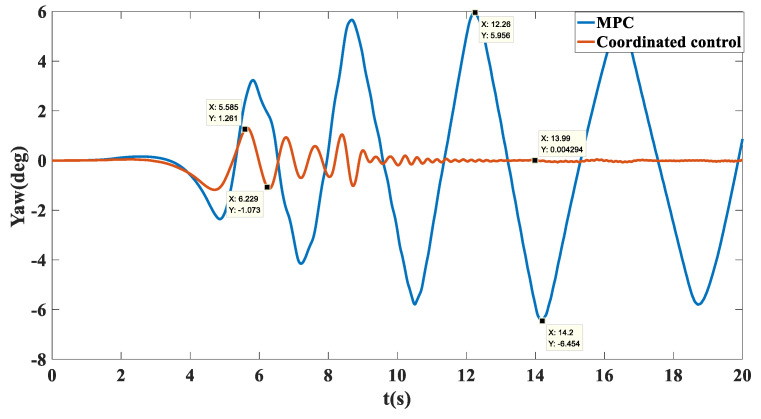
Comparison chart of heading angle tracking on the sine sweep straight road.

**Figure 20 biomimetics-07-00238-f020:**
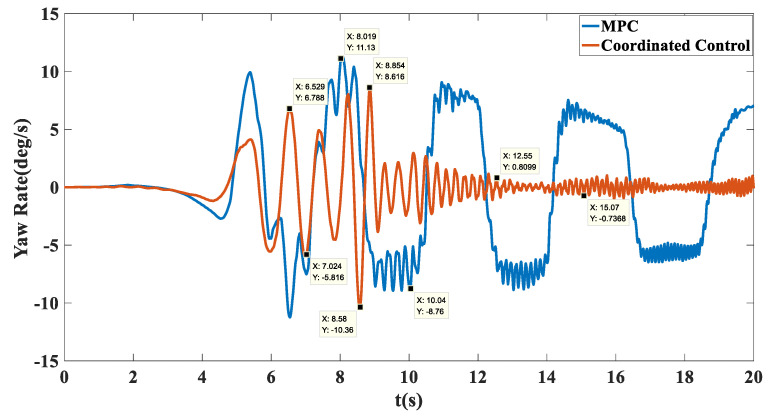
Transverse pendulum angular velocity tracking comparison chart on the Sine sweep straight road.

**Figure 21 biomimetics-07-00238-f021:**
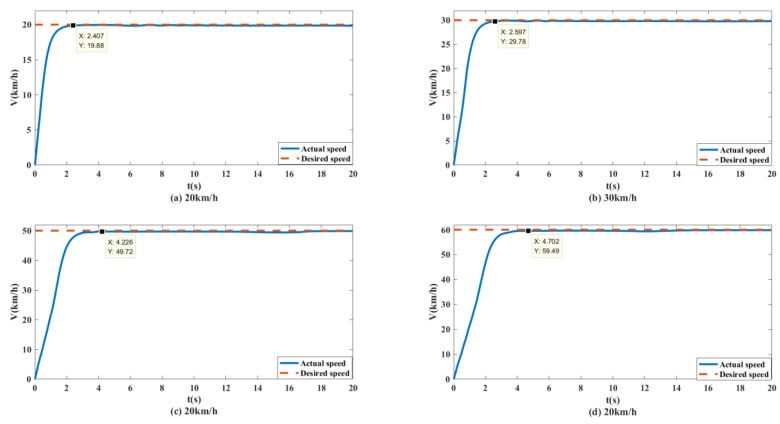
Speed tracking curves of unmanned vehicles at different speeds on the sine sweep straight road.

**Figure 22 biomimetics-07-00238-f022:**
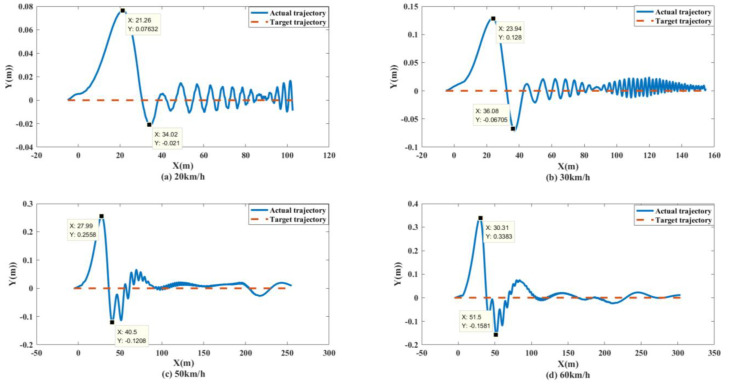
Path tracking curves of unmanned vehicles at different speeds on the Sine sweep straight road.

**Figure 23 biomimetics-07-00238-f023:**
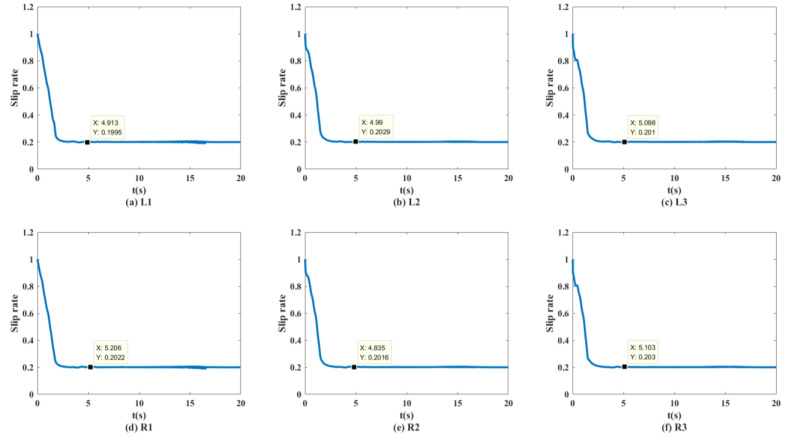
Wheel slip rate of distributed unmanned vehicles at 50 km/h on the sine sweep straight road.

**Table 1 biomimetics-07-00238-t001:** The main parameters of 6WID/4WIS UGV.

Parameters	Values	Units
Sprung mass	2900	kg
Gravitational acceleration	9.8	m/s^2^
The horizontal distance between the center of gravity and the front axle	2	m
The horizontal distance between the front axle and the middle axle	2.2	m
The horizontal distance between the rear axle and middle axle	2.2	m
Wheelbase	2200	mm
Height of center of gravity	1.25	m
Tire diameter	0.996	m
Tire width	0.309	m
Power of in-wheel motor	65	kW
Maximum off-road speed	45	(km/h)
Sampling period	5	ms

## Data Availability

Not applicable.

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
