# Peer review of "Research on Six-Wheel Distributed Unmanned Vehicle Path Tracking Strategy Based on Hierarchical Control"

_biomimetics, 2022, doi:10.3390/biomimetics7040238_

Round 1
Reviewer 1 Report
The method proposed in the manuscript shows a better performance in compared with the conventional MPC. The followings are my opinions about the manuscript:
1. Please supplement the contributions of the method.
2. There are grammar problems in the writing of the manuscript, which makes readers feel challenging. There is an unknown designation in the manuscript, such as the "this problem" in L56 of P2.
3. There are few problems in mathematics in the manuscript. Such as the "t" of Eq10 should be "T", and the matrix S of Eq22 is not a square matrix, but its matrix inversion is used.
4. It is recommended to polish/shorten the manuscript and highlight the key content.
Reviewer 2 Report
1. This paper studies path tracking of the 6-wheel unmanned ground vehicles using an HC framework. Because of the complex kinematic and dynamic characteristics of the 6-wheel vehicles, it is feasible to adopt a hierarchical framework. However, it is common to use MPC and PID as the upper and lower controllers, respectively. It's good to see that the authors conducted the verification on complex road surfaces, such as split and Sine Sweep roads. However, I have the following comments for this work for further consideration.
(1)The title of Section 4 cannot be called Experimental results as it only presents the simulation results verified in Trucksim and Simulink.
(2) The vehicle speed in the verification cases is 40km/h. I'm also interested in the performance of other different speeds, as wide speeds can reflect the adaptability of the control system.
(3) Please clarify the no control and MPC methods in the verification stage. Please clearly explain how to generate the control commands of no control and MPC methods for comparison.
(4) In Line373-377, I think the ideal state of pure rolling discussion is incorrect. The slip rate is non-zero for the driving wheels, even if it is a small value on a high-adhesive road surface.
(5) In Line 296, The optimal slip rate is set as 20%. However, in the subsequent simulation verification, there is no case triggering this extreme situation where 20% is set as the optimal slip rate reference.
(6) How to calculate the slip rate for the ASR controller? The actual vehicle speed is difficult to measure for the 6-wheel driving vehicle; please explain it.
(7) The writing needs to be improved.
Round 2
Reviewer 1 Report
The manuscript is amended according to the previous suggestions. However, I still have few suggestions for the authors:
1. The grammar of the manuscript should be carefully checked again. The first line in page 3 should be in plural form, “The main works are as follows”.
2. The results from different experiments should be shown individually one-on-one in different figures, check out Fig. 21. Figure 21 would deliver a better visual effect by using tracking error instead of the actual velocity.
Reviewer 2 Report
Thank the authors for their efforts to improve the quality of the study. The authors have responded positively to most of my concerns. But I still have a few questions:
1. As shown in Figure 12, the yaw rate curve has a certain oscillation, which means it is not in a stable state of convergence.
2. When the vehicle speeds are 50km/h and 60km/h, there is a large overshoot of the vehicle's lateral displacement, as shown in Figure 14.
A brief discussion of the above two points is preferred. I also suggest addressing the possible future directions of this study in the conclusion section.
In addition, some figures need to be presented more clearly, e.g., Figure 15.
